# The Safety and Efficacy of *Citrus aurantium* (Bitter Orange) Extracts and *p*-Synephrine: A Systematic Review and Meta-Analysis

**DOI:** 10.3390/nu14194019

**Published:** 2022-09-28

**Authors:** Dorottya Koncz, Barbara Tóth, Muh. Akbar Bahar, Orsolya Roza, Dezső Csupor

**Affiliations:** 1Institute of Pharmacognosy, University of Szeged, 6720 Szeged, Hungary; 2Institute for Translational Medicine, Medical School, University of Pécs, 7624 Pécs, Hungary; 3Department of Pharmacy, Faculty of Pharmacy, Universitas Hasanuddin, Makassar 90245, Indonesia; 4Institute of Clinical Pharmacy, University of Szeged, 6725 Szeged, Hungary

**Keywords:** *p*-synephrine, meta-analysis, obesity, food supplements, weight loss

## Abstract

Synephrine has been used to promote weight loss; however, its safety and efficacy have not been fully established. The goals of our study were to give an overview of the safety and efficacy of *p*-synephrine, to systematically evaluate its efficacy regarding weight loss and to assess its safety, focusing on its cardiovascular side effects in a meta-analysis. PubMed, the Cochrane Library, Web of Science and Embase were searched for relevant studies. Only placebo-controlled, human clinical trials with synephrine intervention were included in the meta-analysis. The meta-analysis was reported according to the PRISMA guidelines using the PICOS format and taking into account the CONSORT recommendations. Altogether, 18 articles were included in the meta-analysis. Both systolic and diastolic blood pressure (DBP) increased significantly after prolonged use (6.37 mmHg, 95% CI: 1.02–11.72, *p* = 0.02 and 4.33 mmHg, 95% CI: 0.48–8.18, *p* = 0.03, respectively). The weight loss in the synephrine group was non-significant after prolonged treatment, and it did not influence body composition parameters. Based on the analyzed clinical studies, synephrine tends to raise blood pressure and heart rate, and there is no evidence that synephrine can facilitate weight loss. Further studies are needed to confirm evidence of its safety and efficacy.

## 1. Introduction

The global increase in obesity is strongly connected to modifiable lifestyle factors, including sedentary lifestyle and diet [1]. Obesity is associated with serious health problems and related comorbidities [2,3], and even modest weight loss can result in improved systolic and diastolic blood pressure and blood cholesterol levels [4]. Increased physical activity, a low-calorie diet and the monitoring of body weight can facilitate weight loss in the long term [5].

Dietary supplements are easily available alternatives to medicines; however, data supporting their efficacy are usually scarce, and in some cases, their safety is also questionable [6]. Because of safety concerns, several Active Pharmaceutical Ingredients (APIs), that were considered as effective compounds to support weight loss are no longer available on the market. *p*-Synephrine, a protoalkaloid extracted from the immature fruit or peel of bitter orange (*Citrus × aurantium* L.), is widely used in weight loss and sports performance products [7], yet its efficacy and safety has not been fully established [8].

Synephrine exists in three different positional isomeric forms (ortho, meta and para) [9] (Figure 1). It is generally accepted that only *para*-synephrine (*p*-synephrine) can be found in bitter orange fruits [10,11,12]. Food supplements can contain *meta* (*m*)- and *p*-synephrine, which are both alpha-adrenergic agonists (α-agonists), while the *m*-isoform is the most potent on alpha-1-adrenoreceptors (α_1_-adrenergic receptors) [13]. *Ortho*-synephrine (*o*-synephrine) is not used as a pharmaceutical substance, and its natural occurrence has not been documented [13,14]. Methyl-synephrine HCl (4-HMP, *syn*. oxilofrine HCl) is a prohibited synthetic derivative of *p*-synephrine and has been reported as an adulterant in food supplements [15,16,17,18].

Since the use of ephedrine in food supplements has become prohibited in several European countries and in the USA, *p*-synephrine has gained considerable interest as the main substitute of ephedrine in weight loss products [12,19,20,21]. Synephrine is similar to ephedrine with regard to its structure and mechanism of action; however, it is less lipophilic, resulting in decreased transport through the blood–brain barrier [7,22,23,24]. The use of ephedrine is associated with the increased risk of myocardial infarction, hypertension, and stroke [12,25], but such pronounced effects on the cardiovascular system are not expected when using *p*-synephrine. Animal studies have shown that *p*-synephrine stimulates beta-adrenoreceptors, causing thermogenesis and lipolysis [7], whereas its cardiovascular adverse effects are partly due to its α-adrenergic receptor affinity [26]. Although it is generally acknowledged that *p*-synephrine has milder side effects than ephedrine, its safety and efficacy have not yet been thoroughly studied [8,19].

*p*-Synephrine (hereinafter referred to as synephrine) is used in pre-workout supplements to improve performance and to promote weight loss since it has thermogenic and sympathomimetic properties [27,28]. Synephrine was added to the Monitoring Program in Competitions of the World Anti-Doping Agency [29], but it is not yet considered as a prohibited substance (WADA) in 2022 [30]. However, it is prohibited for use by several professional sporting agencies (the National Collegiate Athletic Association (NCAA), Major League Baseball (MLB), and the National Football League (NFL)) [31].

Currently, there is little if any basis for making definitive statements about the safety of bitter orange extracts or synephrine used in food supplements [9]. Cardiac adverse events, including hypertension, tachyarrhythmia, variant angina, cardiac arrest, QT prolongation, ventricular fibrillation, myocardial infarction, and sudden death, have been the most common adverse effects associated with synephrine intake [27]; however, the prevalence is not known. The average synephrine content of the dried fruit extracts of *Citrus aurantium* has been reported to be between 3% and 6% [9,10,32,33,34]. The French food safety authority (ANSES) concluded in its assessment on synephrine [35] that the intake levels of synephrine from food supplements must remain below 20 mg/day, and it is not recommended to take synephrine in combination with caffeine. It is also recommended to avoid the use of products containing synephrine during physical exercise, and its use by sensitive individuals is discouraged (i.e., people taking certain medications, pregnant or breastfeeding women, children, and adolescents) [35,36]. Because of its known sympathomimetic properties and adrenergic effects on the cardiovascular system, the use of synephrine in food supplements is debated [36]. Even though there is no legislation which limits the content of synephrine and other alkaloids in dietary supplements, based on the Directive 2002/46/EC, each country is supposed to set a maximum level of synephrine [37,38]. There have been reports in the RASFF (Rapid Alert System for Food and Feed) about synephrine, because in some countries, there is a limit regarding its daily dose, and in the reported cases, the products contained more than the maximum [39] (Appendix A).

## 2. Materials and Methods

### 2.1. Literature Search and Selection Criteria

Electronic searches were conducted in the following databases: PubMed, Embase, Web of Science (WoS), and the Cochrane Library. Each database was searched until 17 August 2022. The searching method included the key term synephrine. This meta-analysis of eligible peer reviewed studies was reported in accordance with the PRISMA statement and followed the CONSORT recommendations. Trials were selected if they were (1) human clinical studies, (2) compared known doses of orally used synephrine with a placebo or active control or both, and (3) completed. In order to complete this task, the following PICO (patients, intervention, comparison, outcome) format was applied: P: adults; I: known dosage of *p*-synephrine given per os; C: placebo or control; and O: changes in the body weight, composition, cardiovascular, and metabolic parameters (i.e., heart rate, blood pressure, body weight, body fat, fat mass, fat-free mass, fasting blood sugar level, and RER values). Our hypothesis was that synephrine facilitates weight loss and that its use correlated with cardiovascular adverse effects. This work was registered in PROSPERO (359626). There were no restrictions regarding the number of included patients or the minimal or maximal dosing of *p*-synephrine. The language of the included articles was restricted to English.

### 2.2. Data Extraction and Endpoints

As clarified in the PICO, clinical trials involving adults were included in this meta-analysis. Of these clinical trials, outcomes related to the efficacy and safety of synephrine were extracted. The study endpoints included those values which were present in at least three articles and could be compared. Statistical evaluation could modify the finally analyzed trials/outcomes. The following information from individual studies was extracted: the first author’s name and publication year; the study design and population; the number of participants; other medicines in the intervention group; synephrine regimen; and outcomes.

### 2.3. Quality Evaluations

Two authors (D.K. and D.CS.) performed the literature search. Both authors reviewed the full-text articles and extracted appropriate data from the publications. The risk of bias was analyzed by two of the authors (D.CS. and B.T.), using the Cochrane Risk of Bias Tool, which includes the following domains: random sequence generation, allocation concealment, the blinding of participants and personnel, the blinding of outcome assessment, incomplete outcome data, selective reporting and other scores of bias. For each domain, studies were judged to have a high (red), unclear (yellow), or low (green) risk of bias (Appendix A). Disagreements were resolved by consensus. Risk of bias figures were prepared by using the Review Manager (RevMan) version 5.4.1 software from Cochrane Training site based in London, UK.

### 2.4. Statistical Analysis

An outcome was selected for the final analysis if it was reported in at least three articles; however, further attrition and unique time intervals could modify the analyzed study number/outcome. The assessment of weighted mean difference (MD) and effect sizes (ESs) between test and control group values (synephrine vs. control) was performed as a post–post analysis. A Chi^2^ or Tau^2^ test was used to evaluate the heterogeneity. Deviating study intervention arms were excluded from the final analysis in case there were other extra intervention(s) apart from synephrine (mainly caffeine). In the case of heterogenous subjects based on their caffeine usage, the regular non-caffeine users were selected contrasting to regular high caffeine users. The statistical analyses were conducted using the Review Manager 5.4.1 software. The results were considered statistically significant when the *p* value was less than 0.05.

## 3. Results

### 3.1. Literature Search

The literature search resulted in the identification of 2435 articles (Figure 2). After duplicate removal, the screening of 1472 titles and abstracts resulted in the revision of 51 publications which were retrieved for full-text screening (excluded: *n* = 1421). The 51 full-text articles were assessed for eligibility. After the revision, 21 studies were found to be appropriate for qualitative analysis (excluded: *n* = 30). Articles were excluded if they were not peer reviewed scientific articles (i.e., abstracts or posters) (*n* = 2); were clinical trials but not with synephrine (*n* = 3); were not human clinical studies (*n* = 1); when there were time intervals that were too heterogeneous or did not indicate trial duration (*n* = 5). Further studies were excluded if they had missing statistical values (*n* = 1) or detailed outcomes (*n* = 7), or missing relevant outcomes (*n* = 4), they were not placebo-controlled trials (*n* = 3) or were not clinical studies (*n* = 4). Since one study [40] was open-label and in two studies, the authors did not give the exact dosage of synephrine [41,42], these studies were not eligible for further quantitative analyses. Therefore, a total of 18 studies involving 341 patients were included in the final quantitative analysis. Penzak et al. conducted a two-way, crossover, open-label trial and assessed the cardiovascular effect of approx. 13–14 mg synephrine, which did not significantly alter SBP, DBP, and HR in 12 healthy subjects [40]. Kliszczewicz et al. reported a randomized crossover trial investigating the effects of 100 mg of *Citrus aurantium* (CA) powder (with an unknown dosage of synephrine) compared to caffeine (100 mg) or a placebo. The consumption of *Citrus aurantium* caused a significant time-dependent increase in the HR of the test subjects [41]. In another randomized, double-blind trial, participants consumed 140 mL of a high-energy drink containing unknown dosages of methyl-synephrine [42]. Based on their results, significantly higher SBP was observed in the intervention group during the three-hour study period, but no significant changes were observed in HR or in DBP.

#### 3.1.1. Characteristics of the Trials

The 18 trials included were double-blinded, prospective, parallel or crossover, within-subjects or counterbalanced studies (Appendix A). Seven trials were crossover [43,44,45,46,47,48,49]; five had parallel design [22,50,51,52,53]; four had within-subjects [54,55] or other [56,57] designs; and two were performed in a counterbalanced manner [58,59].

Fourteen trials assessed the acute effects of synephrine [43,44,45,46,47,48,49,53,54,55,56,57,58,59]. In four trials, the effect of a longer treatment duration (a range of 4–8 weeks) was assessed [22,50,51,52]. The daily dosage of synephrine ranged from 6 to 214 mg.

The inclusion criteria of the trials varied from trial to trial. Eight trials only included physically active subjects [22,45,49,50,52,54,56,59]. In eight trials, healthy, normotensive but not resistance-trained subjects were involved [43,44,46,48,51,53,55,57]. Four trials included overweight adults [22,47,52,58]. In several trials, other interventions were also studied, but these study arms were not included in our analysis [45,50,51,53,54,55,56]. Nine trials involved exercise training [22,45,49,50,52,56,57,58,59]. It is important to emphasize that while comparing weight loss and body compositions from the analyzed three articles [22,50,52], all of these trials included exercise intervention, and two involved dietary restrictions [22,52]. Nine trials did not include exercise intervention [43,44,46,47,48,51,53,54,55]. In seven trials, the main analyzed intervention contained substances other than synephrine (e.g., caffeine) [22,44,45,47,50,52,58]. One study [56] claimed that all participants were classified as low caffeine consumers (<50 mg/day), but in the rest of the trials, the authors did not specify whether the subjects were regular or non-regular caffeine users. Bush et al. (2018) [54] and Ratamess et al. (2018) [55] divided groups based on their average caffein consumption. However, the subgroup of regular high caffeine users (>300 mg) was excluded from our final meta-analysis to make the analyzed groups more homogenous. In seven trials, participants were requested to limit caffeine consumption [22,49,51,52,56,57,59].

#### 3.1.2. Demography of the Patients

In the 18 trials analyzed, the common inclusion criterion was the age of 18 years or older, and in every study, the inclusion procedure was in harmony with the Helsinki declaration. In eight cases, pregnant women were excluded [22,43,44,45,46,48,51,52], and also, smoking was an exclusion criterion; only non-smokers participated in nine trials [43,45,49,50,54,55,56,58,59]. The age of the analyzed participants was between 18 and 51 years. Participants were healthy, physically active/exercise trained or in fewer cases, healthy but sedentary and/or overweight. Additionally, more male patients were randomized (68.45%). Taking into consideration the crossover design and the within subject design, 341 subjects were analyzed (Appendix A).

### 3.2. Outcomes

The meta-analyzed outcomes were SBP, DBP, HR, weight loss, body fat percentage, fat mass, fat-free mass, blood glucose, and RER values. In addition, several studies indicated other adverse events, but they were too heterogeneous to be statistically analyzed. These side effects included complaints of headaches [46], hyperventilation [58], racing HR, feelings of dizziness, and feelings of irritability or perspiration [55], palpitations, shortness of breath, nervousness, and blurred vision [50]. Other outcomes were increased VO_2_ uptake [45], increased energy expenditure [44], increased fat oxidation, reduced carbohydrate utilization [57,59], and changes in the ratings of perceived exertion [56,59]. However, the outcomes mentioned above were omitted from the final analysis, since these occurred in less than three articles or were based on other heterogeneous parameters (time intervals) or missing detailed outcomes.

#### 3.2.1. Cardiovascular Parameters

Altogether, 11 trials with 222 subjects and six different time sets and heterogeneous dosages (a range of 10–214 mg) were used to assess the effects of synephrine on blood pressure. The SBP tended to increase in the synephrine group (Figure 3). It was found that 30–45 min after the administration of 20–50 mg of synephrine, the mean difference was 1.21 mmHg (95% CI: −2.57–5.00), which was not significant (*p* = 0.53) but showed a greater effect size for synephrine than for the placebo. After 1 h of the administration of the products containing 49–214 mg of synephrine, the post–post analysis showed a non-significant increase in the SBP (MD 1.56 mmHg, 95% CI: −1.11–4.24, *p* = 0.25). Based on five trials involving 61 subjects, the effect of synephrine (49–180 mg) remained non-significant (MD 3.89 mmHg, 95% CI: −0.99–8.77, *p* = 0.12) after 2 h. After 3 h of the administration, only a slight effect (MD 0.34 mmHg, 95% CI: −2.18–2.87, *p* = 0.79) on the SBP was observable, and it seems that the effect diminished after 6-8 h (MD 0.10 mmHg, 95% CI: −3.82–4.02, *p* = 0.96; dosage 27–108 mg) of consumption. Based on two trials involving 75 subjects, a daily dose of 10–49 mg of synephrine had a significant effect on the systolic blood pressure (MD 6.37 mmHg, 95% CI: 1.02–11.72, *p* = 0.02) after 8 weeks of administration.

The dosages and the included articles were the same in DBP; one difference was that we left one datum point out by 2 h [49], as the statistical SD value “0” could not have been estimated. The administration of synephrine (20–214 mg) did not significantly alter the DBP 30-45 min after administration (MD: −0.88 mmHg, 95% CI: −4.45–2.70, *p* = 0.63); 1 h after administration (MD: −0.89 mmHg, 95% CI: −2.92–1.13, *p* = 0.39); 2 h after administration (MD: 0.48 mmHg 95% CI: −2.22–3.17, *p* = 0.73) (four articles; 49 subjects; 49–108 mg); 3 h after administration (MD: 0.40 mmHg 95% CI: −1.83–2.62, *p* = 0.73); or 6-8 h after administration (MD: −0.43 mmHg 95% CI: −3.52–2.66, *p* = 0.78).

Based on two trials involving 75 participants, a significant effect was observed on DBP after the long-term (8 weeks) usage of 10–49 mg of synephrine (MD 4.33 mmHg, 95% CI: 0.48–8.18, *p* = 0.03) (Figure 4).

Overall, nine trials involving 129 persons with six different time durations and varying dosages of synephrine (range 20–214 mg) studied the effects of synephrine on heart rate. Significant differences between synephrine and the placebo were not reported; however, the heart rate slightly increased after 30 min–6 h from consumption in the synephrine group (Figure 5). It was found that 30–45 min after synephrine consumption (20–50 mg), the mean difference in heart rate was 3.15 beat/min (95% CI: −0.41–6.71, *p* = 0.08); and it was 1.11 beat/min (95% CI: −1.32–3.53, *p* = 0.37) 1 h after the consumption of varying doses (49–214 mg) of synephrine. After 2 h of the ingestion of 49–180 mg of synephrine, a non-significant rise in the heart rate was observed (MD 3.15 beat/min, 95% CI: −0.65–6.96, *p* = 0.10).

Based on four trials involving 43 subjects, 3 h after the consumption of 60–180 mg of synephrine, the synephrine-adjusted increase in heart rate was 3.48 beat/min (95% CI: −0.33–7.29, *p* = 0.07). The effects observed after 4 h (MD 3.25 beat/min, 95% CI: −2.86–9.35, *p* = 0.30) and after 6 h (MD 2.84 beat/min, 95% CI: −2.80–8.48, *p* = 0.32; range 49–108 mg) remained non-significant based on the articles involving these time points.

#### 3.2.2. Weight Loss and Body Composition

The effect of synephrine on weight loss was analyzed in three trials [22,50,52]. The trials lasted for several weeks (42–56 days or 6–8 weeks), and the daily dose of synephrine was 54 mg, 20 mg, and 10 (5 × 2) mg, respectively. Based on our meta-analysis, its effect on weight loss was non-significant (MD 0.60 kg, 95% CI: −5.62–6.83, *p* = 0.85) (Figure 6A). The consumption of synephrine only resulted in a slight decrease in body fat (−1.87%, 95% CI; −3.92–0.18, *p* = 0.07), and the effect was not statistically different from that of the placebo (Figure 6B).

The administration of synephrine (20 or 54 mg) did not result in significant changes in fat mass. The mean difference was −0.32 kg (95% CI: −3.76–3.11, *p* = 0.85) (Figure 6C). A low dosage of synephrine (10 and 20 mg) did not significantly change the fat-free mass based on the included trials (MD 0.47 kg, 95% CI: −4.19–5.13, *p* = 0.84) (Figure 6D).

#### 3.2.3. Other Outcomes

Based on the included trials, blood glucose values were not changed significantly after the administration of 6–103 mg of synephrine. Blood glucose levels changed slightly but not significantly after 2–3 h of the consumption of synephrine (MD 4.62 mg/dL, 95% CI: −3.04–12.29, *p*
*=* 0.24) with moderate heterogeneity (TAU^2^ = 30.32%) (Figure 7 and Appendix A). In the head-to-head studies, the effect of synephrine on blood and plasma glucose [45,54,58] was also not significant.

The acute effects of synephrine (6–60 mg) on the respiratory exchange ratio (RER) were studied in three trials [44,47,58]. The effect of synephrine was not significant after 1, nor after 2, nor after 3 h of administration. After 1 h of consumption, there was no difference (the mean difference was 0.00 (95% CI: −0.03–0.03, *p* = 0.91)), strengthened by a non-classical leave-one-out analysis. After 2 h, synephrine resulted in a −0.02 difference (95% CI: −0.12–0.09, *p* = 0.75), and after 3 h, synephrine resulted in a −0.02 difference (95% CI: −0.12–0.08, *p* = 0.73) (Appendix A).

### 3.3. Risk of Bias

Overall, the methodical quality of the trials included in our final quantitative analysis was reckoned to be acceptable, mostly with low or unclear risks of bias (Appendix A).

The random sequence generation was described in three studies [43,48,49], and the measures taken to ensure allocation concealment were given in eight trials [43,45,49,50,54,55,57,59]; therefore, the selection bias (i.e., random sequence generation or allocation concealment) in these studies was judged to be low. On the other hand, Sale et al. [58] failed to mention whether their study was randomized or not; therefore, it had a high risk of selection bias. In the remaining studies, the information to judge the selection bias was not available; therefore, these studies were reckoned to have unclear risks of selection bias.

Regarding to the performance bias, four studies [43,49,50,59] were judged to have low risks of bias, but in the case of 14 studies, it was not clear whether the treatment and the comparator were identical in size, shape, color, and odor, and who was blinded and until when was not clearly described; therefore, these studies were judged to have unclear risks of performance bias. Four studies had low risks of detection bias [43,46,56,57], but in the other 14 studies, it remained unclear whether the results were assessed in a blinded manner or not.

All the included studies showed low risks of attrition and reporting biases. Five studies had unclear risks of bias, since all of these studies were at least partly founded by pharmaceutical companies related to the studied products, but their influence on the study design, performance, and report are not clearly stated [48,53,54,55,58].

## 4. Discussion

Overall, 18 trials involving 341 adults were analyzed to perform this meta-analysis. Different sets depending on each outcome were applied to assess the weight loss effects of synephrine and to establish its safety based on its effects on cardiovascular variables. Based on the literature, the commonly used doses of *p*-synephrine vary from 25 to 100 mg per day [16,60,61], while our analysis included 6–214 mg of synephrine. Based on our meta-analysis, synephrine did not significantly increase systolic blood pressure acutely, but it significantly increased in the long term (*p* = 0.02) (Figure 3). It had less significant acute effects on diastolic blood pressure and showed similarly significant effects when applied for longer durations (8 weeks) (*p* = 0.03) (Figure 4). After the acute administration of synephrine, the heart rate increased, but the change remained non-significant; the highest increase was measured 3 h after consumption (*p* = 0.07) (Figure 5). Our analysis led to the conclusion that the prolonged use of synephrine did not result in significant alterations in body weight and composition (Figure 6). Based on the analyzed data, the acute administration of synephrine did not change blood glucose and RER values (Figure 7 and Appendix A).

Synephrine’s impacts on cardiovascular health can be predicted by looking at its effects on cardiovascular variables. Since the use of ephedrine is associated with the increased risk of cardiovascular morbidity and mortality, a similarly effective but safe alternative would be greatly appreciated [9,62]. The dosage used during weight loss analysis was 10–54 mg daily for 42–56 days, which nearly coincides with the dosage that resulted in cardiovascular adverse events (increased blood pressure) after 56–60 days after the administration of 10–49 mg of synephrine. Based on our results, the use of synephrine does not lack cardiovascular side effects; therefore, it may not be a safe alternative to ephedrine for those with predisposing comorbidities.

*p*-Synephrine-containing products are marketed for those aiming for higher energy utilization during low- to moderate-intensity exercise [8]. However, it is not yet proven that bitter orange or synephrine consumption can reduce body fat or promote weight loss [8]. Based on our meta-analysis, the prolonged use of synephrine does not result in significant weight loss (*p* = 0.85). A mean difference of 0.6 kg was observed in a total of 94 subjects (47 subjects/intervention with a daily dose of 10–54 mg of synephrine). Synephrine was ineffective in influencing body fat (−1.87%, *n* = 94, *p* = 0.07), fat mass (−0.32 kg, *n* = 69, *p* = 0.85), and fat-free mass (−0.47 kg, *n* = 78, *p* = 0.84).

Some results suggest that *p*-synephrine has the potential to maintain blood glucose levels by stimulating the uptake of excess blood glucose via insulin-dependent or -independent mechanisms in skeletal muscles [63]. Maintaining optimal blood glucose values would be beneficial, suggesting greater hepatic glucose release, which would be favorable during exercise [45,64]. Based on our results, consuming synephrine did not alter blood glucose levels significantly (*p* = 0.24), and blood glucose maintenance was not affected by the consumption of 6–103 mg of synephrine.

Higher RER values indicate that carbohydrates are being predominantly used as fuel, and lower RER values suggest lipid oxidation [65,66]. Physically active and trained subjects exhibit lower RER values than untrained sedentary subjects in response to comparable workloads [67,68,69]. Our analysis included three articles with 33 analyzed subjects overall. In these trials, the effects of synephrine were assessed after 1 h of consumption. After 2–3 h of consumption, data from only 20 subjects were available. Our results suggest that the consumption of 6-103 mg of synephrine does not modify RER values significantly. RER value changes in the studies were presumably related to caffeine, and all three of the analyzed trials administered a supplement which had additional caffeine in it [47].

The strengths of this systematic review with meta-analysis are that only data from published peer-reviewed articles were used, and all the included trials were double-blind trials. However, there are several limitations to the analysis. First, the studied products were different and contained varying doses (6–214 mg) of synephrine with different isometric forms. One study included both *m*- and *p*-synephrine [46], and one study examined the effects of methyl-synephrine HCl [44]. Only six studies included isolated *p*-synephrine [53,54,55,56,57,59]. In the remaining ten studies, the applied products (such as bitter orange extract) contained standardized synephrine [43,48,49] or were mixed with other substances [22,44,45,47,50,52,58]. The main limitation of the study was high attrition, and only a few studies assessed the same outcomes, resulting in relatively small plots. There were no mentions of the the subjects’ ethnicities, which weakens the results, as it is not clear if the groups were diverse enough in this regard. On the other hand, subjects showed a high variety of traits regarding physical activity and age, and more males were analyzed. Some trials included an exercise intervention, which made the data more heterogonous. Previously, Stohs et al. performed a detailed systematic review of synephrine, but there was no meta-analysis on this topic with statistical evaluations; only a literature review was performed [70]. The only available meta-analysis examined oral phenylephrine on nasal airway resistance in patients with nasal congestion, which did not examine *p*-synephrine and did not analyze its weight loss effects [71].

## 5. Conclusions

After the ban of *Ephedra sinica* by the FDA in 2004, there was a clear need for an alternative weight loss supplement with favorable safety profile [12,25]. Synephrine was widely used to promote weight loss, and it is often considered as a safe and effective option, but its safety and efficacy is yet to be studied. When applying synephrine, it should be taken into account that it also has sympathomimetic effects, so it may also increase systolic blood pressure, diastolic blood pressure, and heart rate. Consequently, it may also increase the risk for stroke and myocardial infarction and therefore harm the consumers’ health. Hence, the hemodynamic effects of bitter orange and *p*-synephrine must be established, but unfortunately, studies related to synephrine-induced cardiotoxicity are scarce [27,43].

*p*-Synephrine showed a tendency to increase systolic blood pressure and heart rate acutely, and it significantly increased both systolic and diastolic blood pressure when applied for longer durations. Therefore, our meta-analysis revealed safety concerns related to the use of synephrine. The beneficial effects of synephrine were rather inconclusive. In our meta-analysis, it did not promote weight loss, and neither did it cause beneficial effects on body composition. The overall effects on glucose and RER values were not proved to be favorable. The currently available evidence indicates that synephrine influences blood pressure and heart rate but has no significant effects on weight loss and body composition; therefore, its use is not recommended to promote health. Considering the limitations, it is concluded that further and larger trials are needed to assess the efficacy and safety of synephrine with a lower risk of bias.

## Figures and Tables

**Figure 1 nutrients-14-04019-f001:**
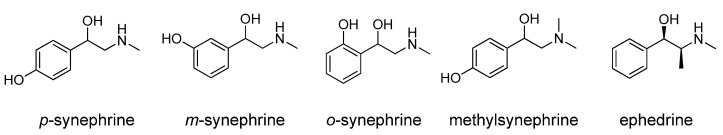
Structures of *para*-, *meta*- and *ortho*-synephrine, methyl-synephrine and ephedrine.

**Figure 2 nutrients-14-04019-f002:**
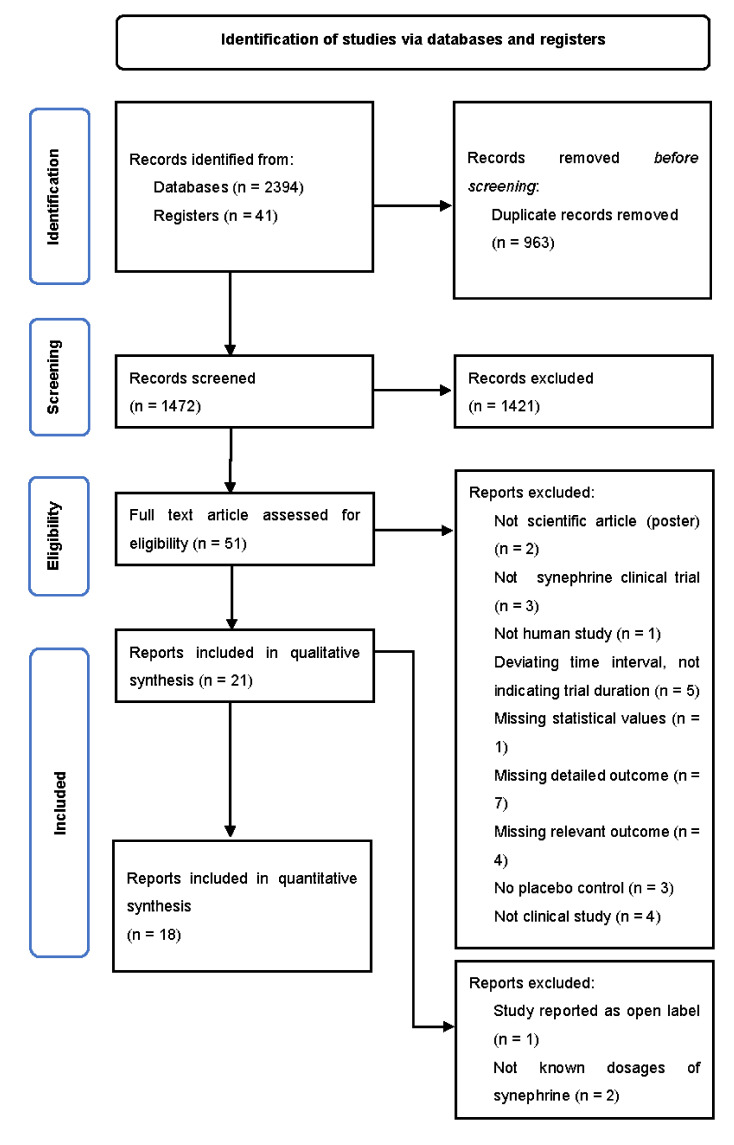
A flow diagram for identification of relevant studies.

**Figure 3 nutrients-14-04019-f003:**
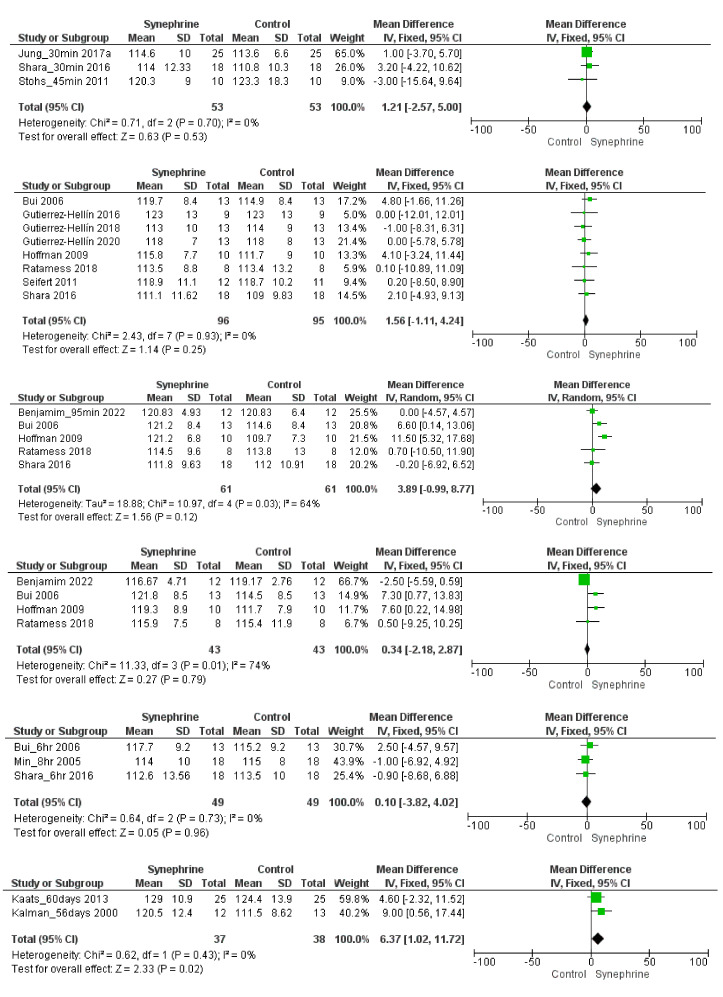
Forest plot diagram of synephrine on systolic blood pressure acutely (after 30–45 min; 1 h; 2 h; 3 h; and 6–8 h) and long-duration (56–60 days) in intervention and control groups.

**Figure 4 nutrients-14-04019-f004:**
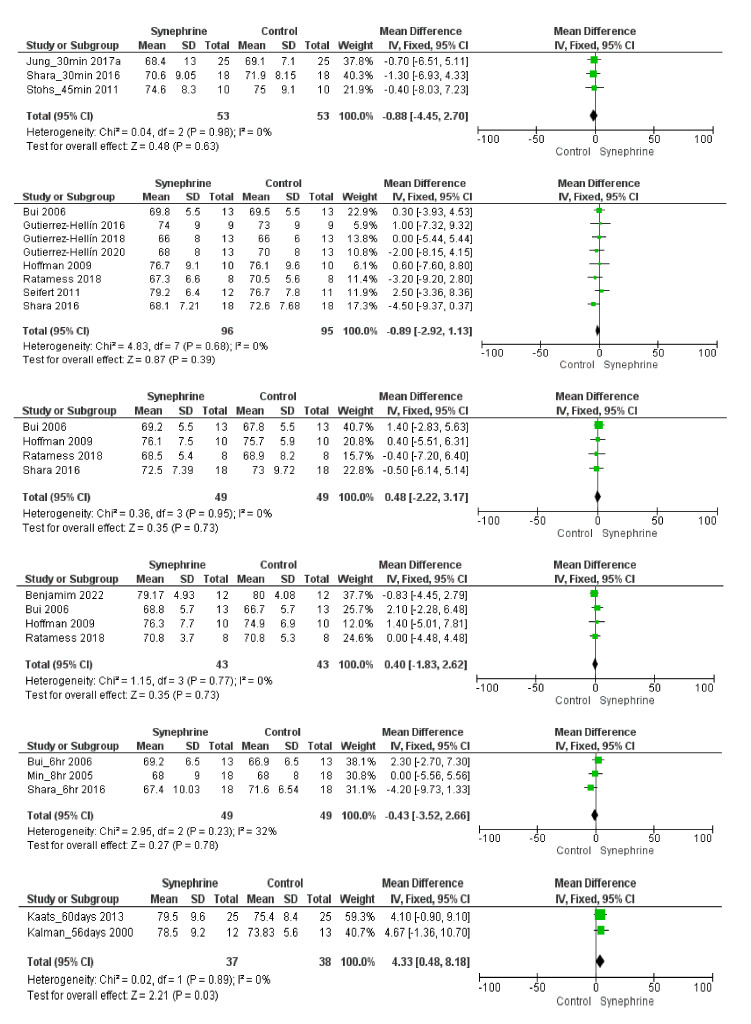
Forest plot diagram of synephrine on diastolic blood pressure acutely (after 30–45 min; 1 h; 2 h; 3 h; and 6–8 h) and long duration (56–60 days) in intervention and control groups.

**Figure 5 nutrients-14-04019-f005:**
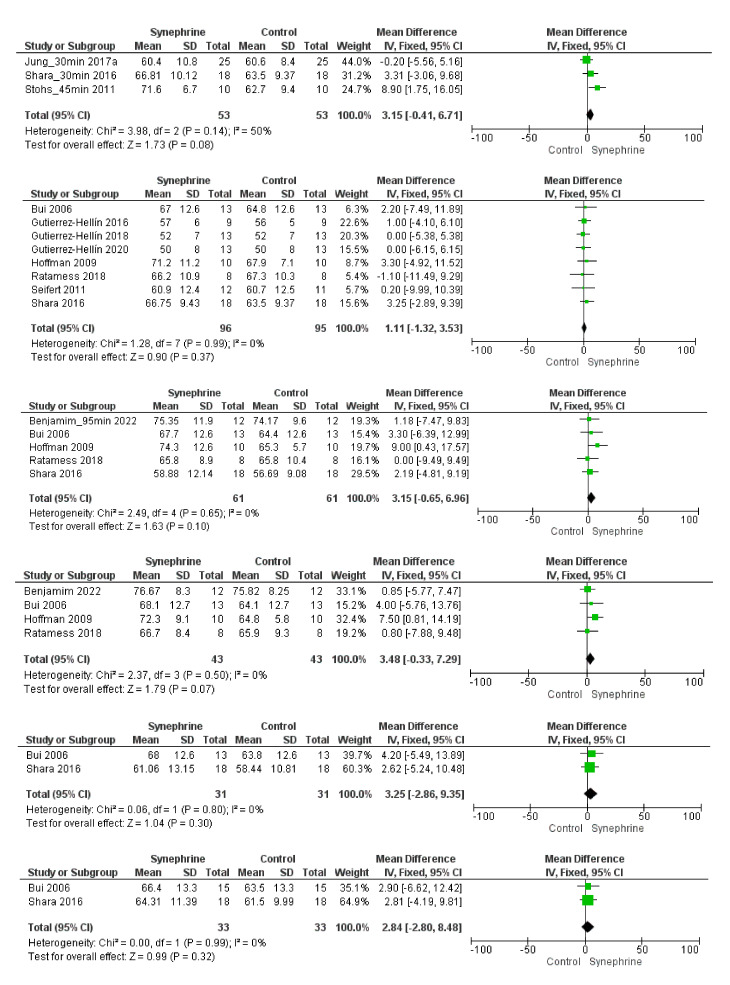
Forest plot diagram of synephrine on heart rate acutely (after 30–45 min; 1 h; 2 h; 3 h; 4 h; and 6 h) in intervention and control groups.

**Figure 6 nutrients-14-04019-f006:**
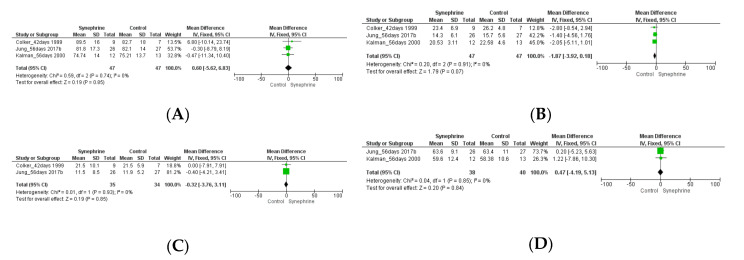
Forest plot diagram of synephrine on weight and body composition from (6–8 weeks) 42–56 days in intervention and control groups: (**A**) weight, (**B**) body fat, (**C**) fat mass, and (**D**) fat-free mass.

**Figure 7 nutrients-14-04019-f007:**
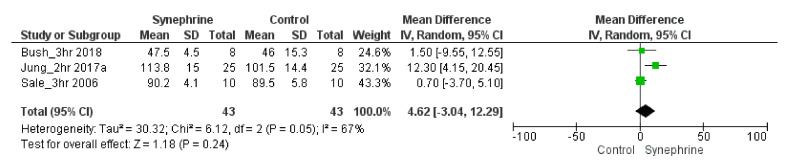
Forest plot diagram of synephrine acutely on blood glucose in intervention and control groups.

## Data Availability

The data presented in this study are available in this article and in the Appendix A.

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
