# Peer review of "The Safety and Efficacy of Citrus aurantium (Bitter Orange) Extracts and p-Synephrine: A Systematic Review and Meta-Analysis"

_nutrients, 2022, doi:10.3390/nu14194019_

Round 1
Reviewer 1 Report
Dear Authors,
The submitted paper for review titled: "The safety and efficacy of Citrus aurantium (bitter orange) extracts and p-synephrine: a systematic review and meta-analysis" was performed in accordance with the rules applied to meta-analysis and meets the requirements for this type of paper by the publisher.
The only objection is the legibility of the figures included in the paper. By their legibility, he recommends the paper for publication.
With best regards
Author Response
Dear Reviewer,
thank you for your comments! We improved the legibility of the figures.
Sincerely, the authors
Reviewer 2 Report
This manuscript provided an overview on the safety and efficacy of p-synephrine and bitter orange extracts on their efficacy on weight loss and their safety focusing on the cardiovascular side effects in a meta-analysis. The authors found out that synephrine tends to raise blood pressure and heart rate and there is no evidence that synephrine can facilitate weight loss.
A few suggestions for the authors.
1. It would be nice if the authors can show the structures of synephrines, methyl-synephrine and ephedrine so that the readers can have a better understanding about their sympathomimetic effects.
2. The review mostly focused on p-synephrine which is a protoalkaloid extracted from the immature fruit or peel of bitter orange. The average percentage of p-synephrine in the bitter orange extract will need to be presented as some of the clinical trials that the authors reviewed used bitter orange extracts.
3. Page 2, line 62-64, the authors claimed “Synephrine was added to the Monitoring Program in Competitions of the World Anti-Doping Agency but not yet considered as a prohibited substance (WADA) in 2009 [29].” It looks like synephrine is still not considered as a prohibited substance in 2022 but some sport organizations have included it on their current list of banned drugs.
Author Response
Dear Reviewer,
thank you for reviewing our paper.
Please find our answers below:
- We have added the structures of the compounds to te manuscript.
- Data were added concerning the synephrine content of bitter orange peel.
- The status of synephrine as a doping agent was discussed more in detail.
We hope that the updated paper meets your expectations and that you find this manuscript suitable for publication.
Sincerely, the authors